# Land Use and Land Cover Influence on Sentinel-2 Aerosol Optical Depth below City Scales over Beijing

**Yue Yang** [1] , **Jan Cermak** [2,3] , **Kangzhuo Yang** [1], **Eva Pauli** [2,3] **and Yunping Chen** [1,*]

1. School of Automation Engineering, University of Electronic Science and Technology of China, Chengdu 611731, China
2. Institute of Meteorology and Climate Research, Karlsruhe Institute of Technology (KIT), H.-v.-Helmholtz-Platz 1, 76344 Leopoldshafen, Germany
3. Institute of Photogrammetry and Remote Sensing, Karlsruhe Institute of Technology (KIT), Englerstr. 7, 76131 Karlsruhe, Germany
* Correspondence: chenyp@uestc.edu.cn

**Abstract:** Atmospheric aerosols can impact human health, necessitating the understanding of their distribution determinants, especially in urban areas. The study discusses the relationships between five major land cover types and aerosol optical depth (AOD) within a city combining the high-resolution satellite-derived AOD products (derived from Sentinel-2) and land cover products (60 m and 100 m, respectively) for Beijing and its surroundings from 2017 to 2019. Contribution analysis is performed to quantitatively evaluate the influences of land cover on regional AOD over the study area. Patterns of aerosol distribution remarkably vary in time and space. Statistics of seasonal average AOD peak in spring and then progressively decline from summer through autumn to winter. High AOD values coincide with a low normalized difference vegetation index (NDVI) and a high normalized difference built-up index (NDBI). Urban and built-up land is a major contributor to regional AOD in the study area, especially in spring; forest and grassland always reduce AOD. Anthropogenic activities have a non-negligible influence on AOD and can even reverse the contribution of a land cover type to aerosols. Insights of the study promote the comprehension of the impacts of land cover on aerosols and air pollution and contribute to the planning of land use within a city.

**Keywords:** aerosol optical depth; land use; land cover; city scale; contribution; Sentinel-2; CGLS-LC100

## 1. Introduction

Atmospheric aerosols are a main stressor of global climate change and exert a profound impact on cloud properties and the land–atmosphere radiative balance [1,2]. Environmentally, aerosols are recognized as one of the primary contributors to urban air pollution and adversely affect human health, by causing, for example, respiratory infection, cardiovascular morbidity, asthma, emphysema, and lung cancer [3–6]. Thus, an understanding of the determinants of aerosol distribution is beneficial as a basis for improving air quality. In a dynamic urban environment, patterns of aerosol distribution in time and space are greatly shaped by land cover types. Land use change is the most remarkable feature of urban development and causes various impacts of anthropogenic activities on the environment [7,8]. The change in land cover not only affects the distribution of emission sources, but also drives regional climate change, thereby impacting the distribution of aerosols [3,9].

Satellite remote sensing enables continuous capture of the aerosol spatial distributions and temporal variations from the regional to the global scale [10]. In the last few years, numerous studies have been carried out on analyzing the influences of land cover on aerosol distribution based on the satellite-derived aerosol products. Xie et al. combined the Moderate Resolution Imaging Spectroradiometer (MODIS) aerosol products and land cover classification derived from Landsat 8 images to examine the impacts of land cover on aerosol

optical depth (AOD) [9]. Sun et al. analyzed the relationship between particulate pollution and land cover for eight representative cities in China with the MODIS aerosol products and land cover products (MOD04 and MOD12, respectively) [11]. Liu et al. used long-term MODIS AOD products and land cover type products (MOD04 and MCD12C1, respectively) to quantify the contributions of six major land cover types to AOD [8]. Feng et al. separated the contributions of climatic and surface to aerosols and then estimated the actual surface influence on atmospheric aerosols relying on a multilinear regression model [12].

Few studies have focused on the relationships between land cover types and aerosols below city scales, since most aerosol products are limited to coarse spatial resolution (several to tens of kilometers). Given the spatial heterogeneity of land cover, more spatial detail in aerosol would be desirable. Hence, further studies of the effects of land cover on aerosols below city scales are of great scientific interest. The purpose of this study is to characterize the spatiotemporal distributions of AOD and to explore the contributions of five major land cover types to aerosol distribution, with the high-resolution Sentinel-2 AOD products and Copernicus Global Land Service land cover products (CGLS-LC100) (60 m and 100 m, respectively). Because built-up area and vegetation density are two major factors of land cover, the relationships between vegetation coverage, building density (approximated by normalized difference vegetation index (NDVI) and normalized difference built-up index (NDBI), respectively) and aerosol are also investigated.

The study is structured as follows: Section 2 details with the study area and data sets. Section 3 introduces the processing methodology. Section 4.1 discusses the spatiotemporal patterns of AOD for the study area. The relationships between AOD, NDVI, and NDBI are investigated in Section 4.2. Section 4.3 analyzes the influences of land cover types on AOD. The contributions of land cover types to regional AOD are further presented in Section 4.4. The conclusions are provided in Section 5.

## 2. Materials

### 2.1. Study Area

The study focuses on a small rectangular area (equal to the area covered by a Sentinel-2 image) around Beijing, China (Figure 1). Beijing is the capital of China with marked aerosol pollution, heavy industrialization, complex surface conditions, and a dense population of more than 10 million [13]. Central Beijing is a typical urban area with dense vegetation surrounding the west, northwest, and northeast, and agricultural farmlands to the southeast. The dominant land cover types in the study area are forest and urban and built-up land, followed by cropland, grassland, and sparsely scattered open shrubland. Thus, the area is suitable for understanding and quantifying the influences of land cover on AOD. In this study, analysis is carried out on five main land cover types, i.e., cropland, grassland, forest, urban and built-up land, and open shrubland. Barren land is excluded from the analysis to avoid statistical inaccuracies resulting from the small area concerned.

### 2.2. Aerosol Data

AOD is an important indicator of aerosol loading, defined as the extinction of radiation in an atmospheric column [14]. In this study, AOD is used as the proxy for aerosol presence. Furthermore, high-resolution (original resolution of 60 m) AOD retrieved from Sentinel-2 measurements covering three years (2017−2019) are selected. The products are retrieved by application of the Yang et al. algorithm, which combines surface reflectance correlations and temporal signatures over the vegetated areas and bright areas to generate AOD [10]. Sentinel-2 AOD has been well validated and shown to have a good performance over complex urban areas, with a correlation coefficient of 0.927 against Aerosol Robotic Network (AERONET) measurements. Compared with the widely used MODIS AOD products (i.e., MOD04_L2 and MCD19A2), Sentinel-2 AOD from the algorithm is superior in both spatial resolution and accuracy. In order to match the resolution of the obtained land cover maps, we downsample the Sentinel-2 AOD products to 100 m using the nearest-neighbor (NN) method; NN is chosen here to be consistent with the resampling method used in

Yang et al.'s research. Further information on quality and availability of the used Sentinel-2 AOD products can be found in Yang et al. [10].

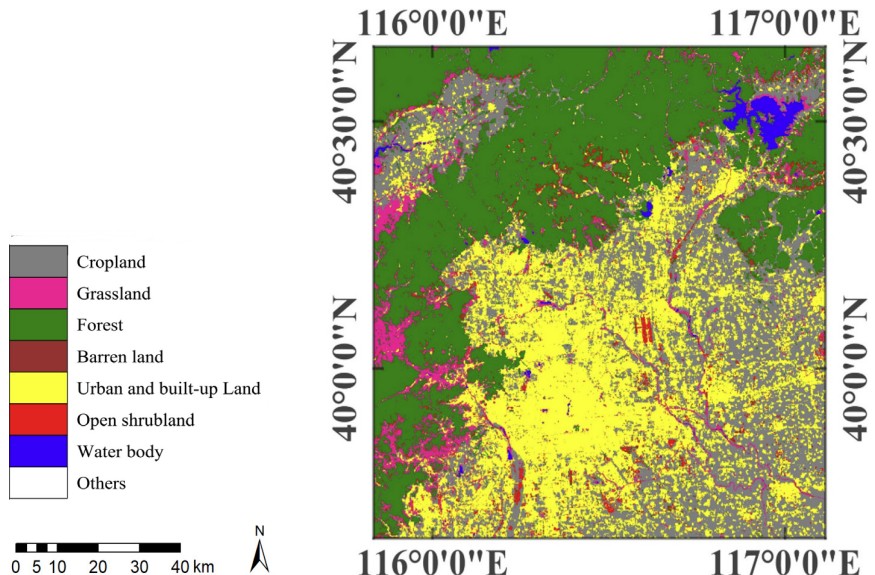

**Figure 1.** Land cover of the study area in 2017.

*2.3. NDVI and NDBI*

In a typical urban environment, land cover is significantly determined by the built-up area and vegetation cover. In order to better comprehend the influences of land cover on aerosols, the interactions between AOD, degree of built-up areas, and vegetation density are discussed. Hereinafter, NDVI is used to approximate vegetation coverage, whereas NDBI serves as the proxy of building density. Sentinel-2 Level-1C (L1C) products (downloaded at https://earthexplorer.usgs.gov/, accessed on 31 August 2021) with the same overpass time of the collected AOD products are obtained to calculate the NDVI and NDBI. Using the nearest-neighbor method, both indices are then resampled to 100 m to match the resolution of the acquired land cover maps.

NDVI quantifies vegetation and reflects its state by measuring the difference between near-infrared (NIR) and red channels as:

$$\text{NDVI} = \frac{\rho_{0.842} - \rho_{0.665}}{\rho_{0.842} + \rho_{0.665}} \tag{1}$$

where $\rho_\lambda$ represents the Sentinel-2 reflectance at wavelength $\lambda$ (µm). The NDVI values given by Equation (1) range from $-1$ to 1, with vegetation coverage increasing from non-vegetated to densely vegetated [15].

NDBI takes advantage of the shortwave infrared (SWIR) and NIR channels to emphasize the manufactured built-up areas and quantify their density as:

$$\text{NDBI} = \frac{\rho_{1.610} - \rho_{0.842}}{\rho_{1.610} + \rho_{0.842}} \tag{2}$$

The NDBI values also range from $-1$ to 1, with higher values corresponding to increased building density. A positive value of NDBI represents urban land areas, whereas a negative value of NDBI stands for non-urban land areas [16].

*2.4. Land Cover Data*

The CGLS-LC100 maps physical coverage of the Earth's surface annually with a spatial resolution of 100 m from 2015 to 2019 [17], accessed via https://lcviewer.vito.be/ download on 24 November 2021. The discrete classification map layer, included in the CGLS-LC100

version (V) 3.0 product, supplies 23 discrete land cover classes under the definition of the Land Cover Classification System (LCCS) scheme developed by the United Nations (UN) Food and Agriculture Organization (FAO). Herein, the discrete classification map layers in the CGLS-LC100 V3.0 products for the years 2017 to 2019 are selected and reclassified into eight land cover types (Table 1) under the instruction of Liu et al. [8]. Analysis of the influences on AOD is then performed on the five main land cover types, i.e., cropland, grassland, forest, urban and built-up land, and open shrubland.

**Table 1.** Original Labels and Aggregation of Copernicus Global Land Service Land Cover Product (CGLS-LC100) Classes Used in the Study.

| Code | Label | Original Labels Included |
|:---:|:---:|:---:|
| 1 | Cropland | Cultivated and managed vegetation/agriculture |
| 2 | Grassland | Herbaceous vegetation, herbaceous wetland, moss, and lichen |
| 3 | Forest | Closed forest (evergreen needle leaf, deciduous needle leaf, evergreen broad leaf, deciduous broad leaf, mixed, unknown), open forest (evergreen needle leaf, deciduous needle leaf, evergreen broad leaf, deciduous broad leaf, mixed, unknown) |
| 4 | Barren land | Bare/sparse vegetation |
| 5 | Urban and built-up land | Urban/built up |
| 6 | Open shrubland | Shrubs |
| 7 | Water body | Permanent water bodies, open sea |
| 8 | Others | No input data available, snow and ice |

## 3. Methodology

The AOD products and land cover type products are used from 2017 to 2019. The study focuses on four main aspects: (1) the spatiotemporal patterns of AOD for the study area; (2) the relationships between AOD, NDVI, and NDBI; (3) the impacts of the five main land cover types on AOD; and (4) the contributions of land cover to regional AOD.

Monthly, seasonal, and annual mean AOD images are averaged from all Sentinel-2 AOD products over a month, a season, and a calendar year, respectively. Three-year mean AOD, NDVI, and NDBI images are acquired by averaging all the corresponding images over the study period (2017−2019). In order to reduce the statistical uncertainties in calculated monthly, seasonal, annual, and three-year mean values, pixels with less than two observations available during the calculation period are removed. Averaging AOD values for all pixels in the study area over a calendar year generates the annual mean AOD value.

Liu et al. proposed that the contribution of AOD from a land cover type to the regional AOD can be estimated from [8]:

$$C_i = S_i \times dT_i \tag{3}$$

Where $S$ is the percentage of a given land cover type; $dT$ is the difference between the average AOD for a given land cover type and the average AOD for the entire area; footer $i$ represents the land cover type; and $C$ is the estimated contribution. Dividing the area of a given land use type by the area of the entire study area obtains the percent $S$. Averaging AOD values for all the pixels over the given land cover type and the entire study area yields the land cover type mean AOD and the regional mean AOD, respectively. A positive $C$ reports that the given land cover type favors the regional AOD, whereas a negative $C$ indicates the opposite. The higher the value $C$, the greater positive effect of a given land use type on the regional AOD.

## 4. Results and Discussion

### 4.1. Spatiotemporal Characteristics of AOD

The spatial patterns of annual average AOD from 2017 to 2019 for the study area are illustrated in Figure 2. High AOD values are found mainly in the central urban areas and a small area in the northwest of the study area, where urban and built-up land and cropland dominate. AOD values for areas covered mostly by forest and grassland are low, with most of them under 0.5. The spatial differences in AOD distribution are consistent with an earlier study and signal the influence of land cover on the spatial patterns of AOD [18].

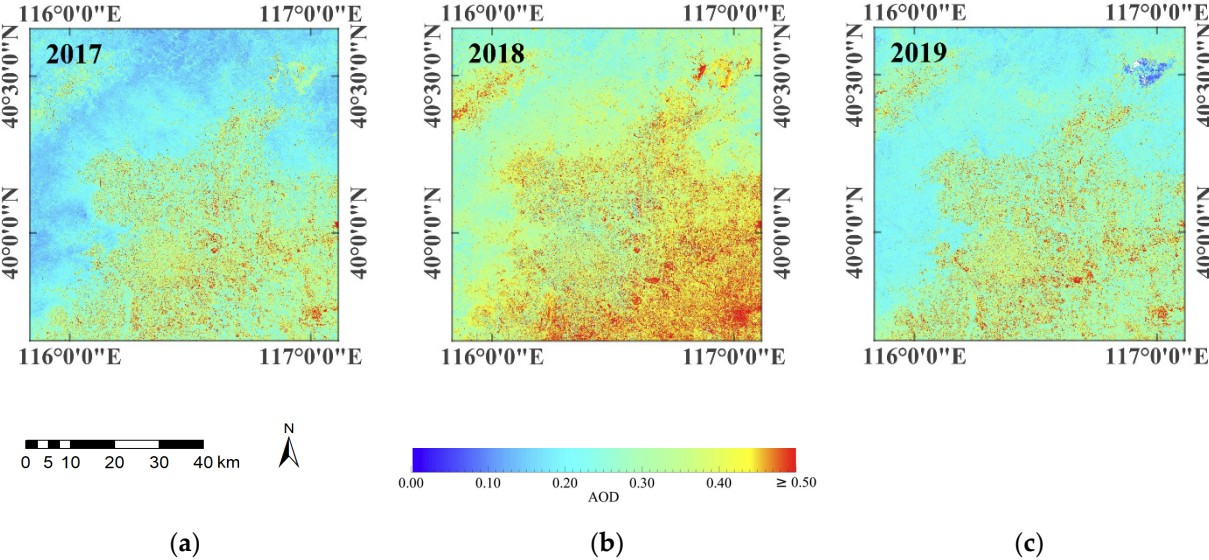

**Figure 2.** Spatial distributions of the annual average aerosol optical depth (AOD) for the study area in (**a**) 2017, (**b**) 2018, and (**c**) 2019.

High AOD values for 2018 are observed mainly in the Southeastern area, around Langfang city, Hebei Province, and decrease with distance from it. The observation suggests that neighboring areas have a non-negligible influence on local aerosol distributions. The annual average AOD for year 2018 (0.365) is higher than that for year 2017 (0.264) and 2019 (0.289), attributed to the low and non-uniform temporal resolution ($\geq$10 days) of Sentinel-2 AOD products. We note that the applied Sentinel-2 aerosol retrieval algorithm has strict cloud filtering (land cloud cover < 10%) and will not run on non-eligible products to ensure the retrieval accuracy.

Figure 3 shows the statistics of three-year averaged monthly mean AOD for the study area. The monthly mean AOD is found to peak in spring (March and April), whereas it touches the bottom in winter (December and January). Jiang et al. concluded that high AOD in spring are mainly attributed to intensive and frequent dust storms in the Beijing-Tianjin sand source region [19]. In addition, biomass burning and less dense vegetation also contribute to the high spring AOD [20]. June and September also have high AOD values. Abundant atmospheric water vapor linked to precipitation in summer may be responsible for the high AOD values in June [21]. Moisture in the atmosphere leads to the hygroscopic growth of particles (aerosol swelling), enhancing the scattering coefficient and thereby increasing the AOD values [22]. Furthermore, the high temperature in summer stimulates photochemical reactions and accelerates soil drying, resulting in increased aerosol loading [23–25]. Harvesting possibly leads to high AOD in September. The climate in Beijing in winter is dry and cold, condensing photochemical reactions, thereby decreasing the AOD values in this season.

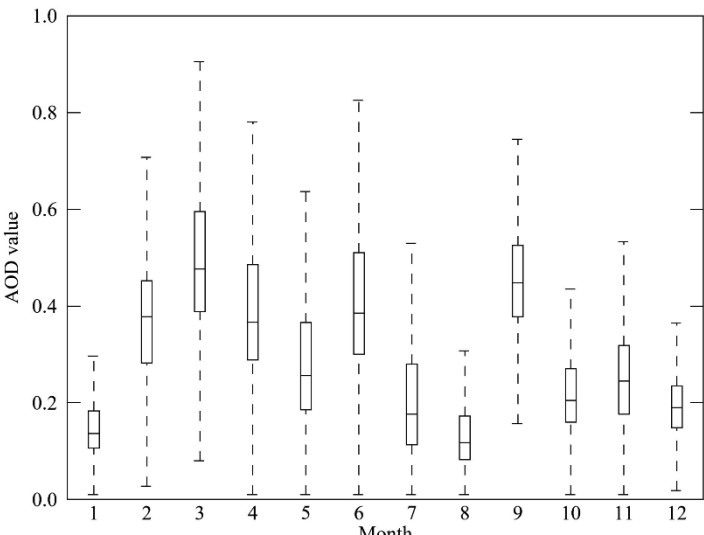

**Figure 3.** Temporal pattern of monthly mean AOD values. Boxes show the lower to the upper quartile value, with a black line at the median. Whiskers extend to most extreme mean AOD values within a month.

The spatiotemporal patterns of AOD illustrate that (1) the central urban area has higher AOD values than the surrounding rural area; (2) local aerosols are influenced by regional air quality; (3) spring is the most polluted season for the study area; and (4) anthropogenic activities, meteorological factors, land cover types, and other factors impact aerosol distributions. In this study, we focus on land cover.

### 4.2. Relationships between AOD, NDVI, and NDBI

Land cover is closely associated with the NDVI and NDBI and can be derived from the two indexes [11,26]. Therefore, the NDVI and NDBI are employed here to preliminarily investigate the relationships between AOD and land cover types. Specifically, AOD−NDVI relationship statistically model the correlation of AOD to vegetation coverage; the AOD−NDBI relationship presents the impact of building density on AOD (Figure 4). With increasing NDVI, AOD is shifted toward lower values, displaying a negative correlation between the vegetation density and aerosol concentration. The dependence of AOD on NDBI is opposite, with a high AOD value that coincides with a high NDBI. The NDVI values are found mainly between 0.0 and 0.5; the NDBI values are mainly clustered between −0.15 and 0.15. The determination coefficients ($R^2$, calculated from the Pearson correlation coefficient) of AOD−NDVI and −NDBI relationships are of 0.591 and 0.612, respectively, revealing a reasonable correlation between AOD and NDVI/NDBI. Some research also characterized the links between NDVI/NDBI and AOD amount; however, they only found weak connections [3,9]. One possible reason behind this failure could be coarse resolution (≥1 km) of the AOD products utilized in these studies. The method of sorting and averaging AOD by NDVI/NDBI bins instead of the AOD−NDVI or −NDBI pixel pairs employed by previous research also greatly contributes to the findings of reasonable correlations. Obvious nonlinearity is observed in both AOD−NDVI and −NDBI relationships, due to the contributions of other factors (for example, meteorological factors, topography, and anthropogenic activities).

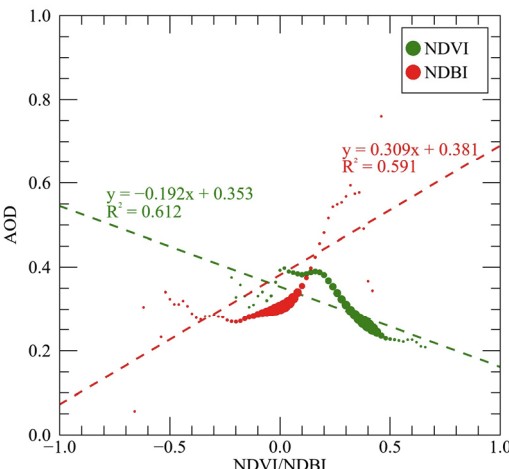

**Figure 4.** Relationships between AOD and normalized difference vegetation index/normalized difference built-up index (NDVI/NDBI). Three-year average AOD values are sorted by corresponding NDVI or NDBI values and averaged into bins. Bubble represents a NDVI or NDBI bin (from −1 to 1, with an interval of 0.02), with size indicating the number of available AOD retrievals in this bin. The x-axis value of each bubble refers to the maximum NDVI or NDBI value of the bin, and the y-axis value refers to the AOD mean of the bin.

NDVI interacts with NDBI. Thus, the dependency of AOD on combined NDVI and NDBI is further emphasized in Figure 5. The contour can be coarsely divided into three areas: red area on the top, yellow area on the left, and blue area on the right. The blue area, dominated by low AOD values, highlights the negative effect of low to moderate NDBI and high NDVI on AOD. In contrast, the red area exhibits a strong contribution of moderate NDVI and high NDBI to high AOD in the study area. The influence of low to moderate NDVI and NDBI on AOD (the yellow area) is complicated, suggesting that more AOD determinants, i.e., meteorological factors, should be taken into account to explain the AOD distributions over sparse vegetated and non-urban areas in full. In conclusion, high NDVI combined with low NDBI is more likely to lead to a low AOD value. Equally, high AOD tends to occur in areas with low NDVI and high NDBI values.

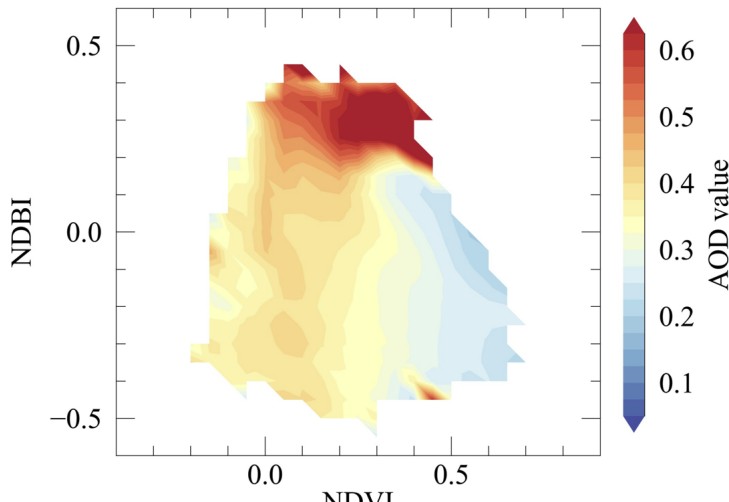

**Figure 5.** Correlations between AOD and combined NDVI and NDBI values.

Results presented in this section indicate that aerosol loading follows built-up area fraction and vegetation density. An efficient way to improve air quality is simultaneously to increase the dense vegetation coverage and decrease the building density.

### 4.3. Impacts of Land Cover on AOD

To further explore the relationships between land cover and AOD, we divide AOD values into two groups: ≤0.5 and >0.5 under the instruction of [27,28]. Percentages of the AOD (seasonal AOD average image) groups for different land cover types in different seasons are shown in Figure 6. AOD values for the study area are dominated by values under 0.5, with domination increasing from spring (MAM), summer (JJA), autumn (SON) to winter (DJF), consistent with Li et al. [29]. The percentage of high AOD values in spring is far greater than in any other season, probably pointing to spring dust storms, biomass burning, or the formation of secondary aerosols. Compared with the grassland and forest, urban and built-up land, open shrubland, and cropland are more susceptible to dust pollution, with a markedly higher >0.5 percentage in spring (Figure 6a).

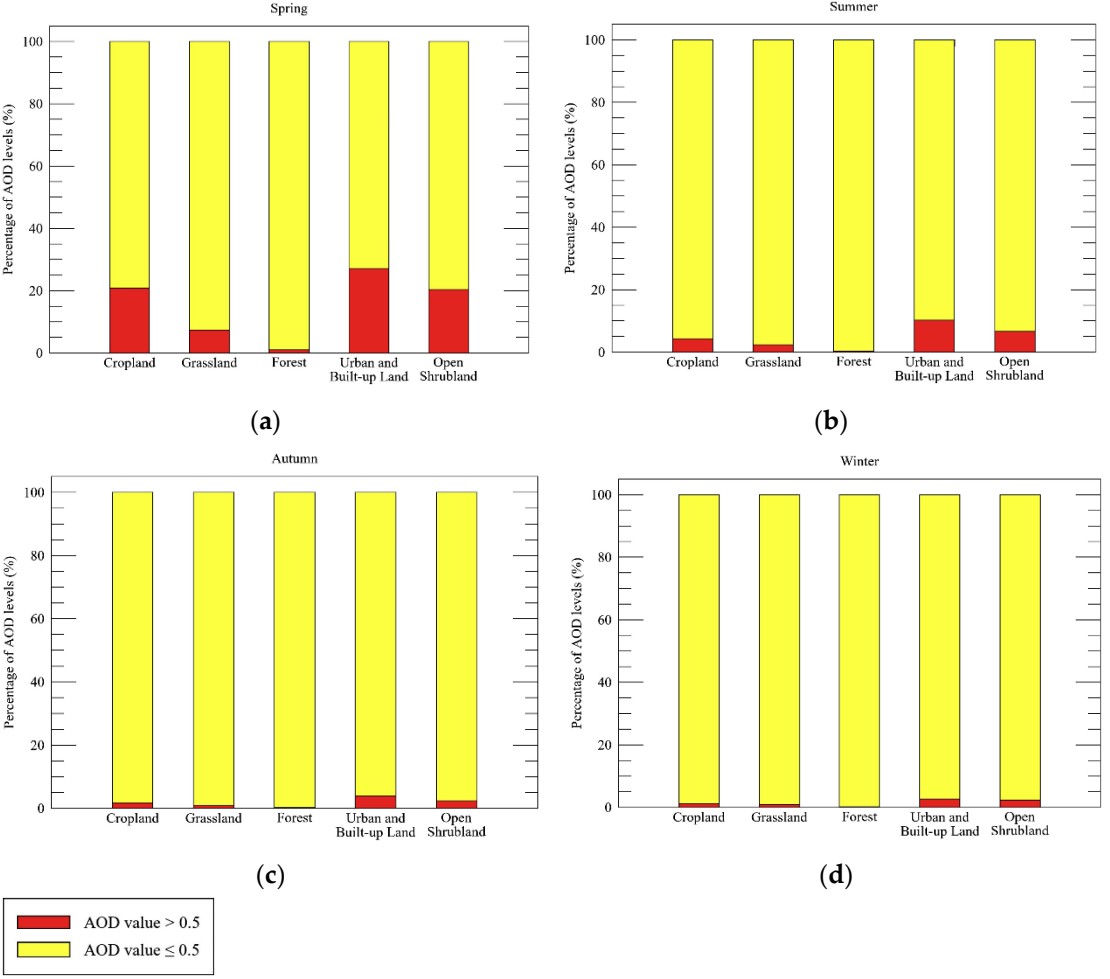

**Figure 6.** AOD group percentages for different land cover types in (**a**) spring, (**b**) summer, (**c**) autumn, and (**d**) winter.

The fraction for the five main land cover types in the >0.5 group, from large to small, are as follows: urban and built-up land > open shrubland > cropland > grassland > forest. A different arrangement of high AOD value (>0.5) percentages between cropland and open shrubland are found in spring, with higher percentage of AOD > 0.5 shown in cropland. The dominant crop types are wheat and maize in Beijing, and the sowing dates are always in or around April [8,30,31]. The booming sowing activity is responsible for the high cropland AOD in spring.

The differences in AOD group percentages for different land cover types describe that land cover types strongly modulate AOD distributions. Land cover type with infrequent

anthropogenic interference and dense vegetation is more possibly accompanied by low AOD values.

### 4.4. Contributions of Different Land Cover Types to AOD

The study quantitatively estimates the contributions of different land cover types to regional AOD over the study area using Equation (3). Table 2 shows the estimated contributions of the five main land cover types and the differences between average AOD for a land cover type and average AOD for the entire study area. There was only one Sentinel-2 AOD product available in summer 2018. To avoid statistical errors, *dT* and *C* for summer 2018 are not provided here.

**Table 2.** Contributions of the Five Land Cover Types to AOD in the Study Area.

| Land Cover Type | Year | S (%) | Spring | | Summer | | Autumn | | Winter | |
|---|---|---|---|---|---|---|---|---|---|---|
| | | | *dT* | *C* | *dT* | *C* | *dT* | *C* | *dT* | *C* |
| Cropland | 2017 | 27.638 | 0.040 | 1.114 | 0.023 | 0.646 | 0.008 | 0.233 | 0.024 | 0.672 |
| | 2018 | 27.533 | 0.048 | 1.334 | / | / | 0.037 | 1.021 | 0.020 | 0.544 |
| | 2019 | 27.455 | 0.037 | 1.009 | −0.015 | −0.418 | 0.017 | 0.471 | 0.019 | 0.514 |
| Grassland | 2017 | 5.950 | −0.045 | −0.269 | −0.047 | −0.279 | −0.010 | −0.062 | −0.022 | −0.130 |
| | 2018 | 5.968 | −0.016 | −0.094 | / | / | −0.009 | −0.053 | −0.014 | −0.081 |
| | 2019 | 5.918 | −0.021 | −0.122 | −0.032 | −0.190 | −0.022 | −0.129 | −0.025 | −0.149 |
| Forest | 2017 | 34.263 | −0.124 | −4.263 | −0.066 | −2.245 | −0.052 | −1.786 | −0.064 | −2.196 |
| | 2018 | 34.240 | −0.060 | −2.067 | / | / | −0.061 | −2.104 | −0.043 | −1.482 |
| | 2019 | 34.223 | −0.088 | −3.025 | −0.033 | −1.127 | −0.062 | −2.130 | −0.038 | −1.293 |
| Urban and Built-up Land | 2017 | 28.210 | 0.120 | 3.381 | 0.070 | 1.964 | 0.056 | 1.567 | 0.044 | 1.230 |
| | 2018 | 28.241 | 0.028 | 0.794 | / | / | 0.038 | 1.077 | 0.030 | 0.861 |
| | 2019 | 28.299 | 0.074 | 2.092 | 0.061 | 1.729 | 0.062 | 1.768 | 0.027 | 0.766 |
| Open Shrubland | 2017 | 2.829 | 0.036 | 0.103 | 0.006 | 0.018 | 0.016 | 0.045 | 0.010 | 0.027 |
| | 2018 | 2.802 | 0.032 | 0.088 | / | / | 0.019 | 0.054 | 0.018 | 0.050 |
| | 2019 | 2.768 | 0.036 | 0.098 | 0.008 | 0.023 | 0.009 | 0.025 | 0.015 | 0.042 |

Grassland and forest always have negative *dT*, interpreting that the average AOD of the two land cover types is lower than the regional AOD average. The negative *dT* also implies that both forest and grassland function as aerosol decreasers due to the absorption, blocking, and deposition of particles by vegetation. Multiplied by the corresponding land cover type percentage, the contributions of the two land cover types are as follows: forest < grassland, demonstrating that forest has a more pronounced decreasing function on regional aerosols than grassland.

Positive values of *dT* are mostly driven by cropland, urban and built-up land, and open shrubland. However, the open shrubland is supposed to have a negative value of *dT*, since earlier studies suggested that the open shrubland serves as the vegetation barrier against aerosols [8,32]. Unfortunately, the small area (*S* < 3%) and scattered pattern of open shrubland masks this effect, resulting the positive values of *dT* and *C*. Overall, urban and built-up land positively contribute most strongly to regional AOD, followed by cropland and finally open shrubland.

Maximum and minimum *C* are observed in spring, for urban and built-up land and forest, respectively. The dust storms occur in spring in the study area. The observations show that urban and built-up land are strongly susceptible to dust storms. Moreover, the complex composition of aerosols and heavy industrial emissions in urban and built-up land may also partially contribute to the maximum *C*. The minimum *C* reflects that forest is very efficient at weakening aerosol pollution due to deposition. Dense vegetation can also suppress the aerosol dispersion by reducing wind speed.

During summer, the *dT* value of cropland changes from positive to negative, from 2017 to 2019, thereby leading to a change in the direction of contribution *C*. The abnormal

change presented here explains a contradictory function of cropland on aerosols. The 2019 world horticultural exposition held in Yanqing District, Beijing (Expo 2019 Beijing) may be responsible for the negative contribution. Yanqing District is in northwest Beijing, in the cropland region seen in the upper left of Figure 1. To cater the theme—"Live Green, Live Better", Expo 2019 Beijing did great environmental friendly efforts, including adopting a 100% green transportation mode in the Expo 2019 park, constructing low-energy buildings, making full use of renewable energy, and transforming the Expo 2019 park into a regional large-scale (area of approximately 5.03 km$^2$) ecological park after the conference (cf. "International Horticultural Exhibition 2019 Beijing China", Construction21, 30 July 2019). In return, artificially reversing the contribution value *C* of cropland from positive to negative.

In summary, the variations in *dT* and *C* quantify a close relationship between regional AOD and land cover types, with grassland and forest consistently reducing regional AOD, whereas cropland, urban and built-up land, and open shrubland being positive contributors to regional AOD. Particularly, the variations in AOD of different land cover highlight the importance of anthropogenic agency on regional aerosols: high AOD observations coincide with frequent anthropogenic interferences. The results presented in this section also imply that the landscape pattern of land cover impacts aerosols.

## 5. Conclusions

Satellite-derived AOD and land cover data for Beijing and its surrounding areas from 2017 to 2019 are used to explore the influences of land cover types on atmospheric aerosols below city scales. The study characterizes the spatiotemporal patterns of aerosol for the study area. The AOD distributions exhibit obvious annual and seasonal variations. In addition, local aerosols are impacted by regional air quality. Compared with the surrounding rural areas, central urban areas have a higher AOD.

Investigations are also performed to the relationships between AOD, NDVI, and NDBI, and suggest that both NDVI and NDBI are correlated to AOD. However, NDVI or NDBI alone cannot explain the aerosol distribution. In combination, land cover types with prevailing high NDBI and low NDVI values have a higher probability to cause high AOD, whereas land cover dominated by low NDBI and high NDVI values tends to cause low AOD. Vegetation is of great importance for decreasing AOD; urbanization increases the aerosol pollution instead.

In terms of the influences of the five major land cover types on AOD, the urban and built-up land is most likely to feature high AOD, followed by open shrubland and cropland. In contrast, grassland and forests are limited to lower AOD. For environmental policy makers, the three pollution-susceptible areas, i.e., urban and built-up land, open shrubland, and cropland, worth priority, especially in spring.

The study quantifies the contributions of different land cover types to the regional AOD for the study area. Generally, land cover types affected by anthropogenic activities (i.e., cropland, urban and built-up land, and open shrubland) are positive contributors to regional AOD. Grassland and forest, dominated by dense vegetation, consistently help to alleviate regional AOD. Maximum and minimum contributions are found in spring, for urban and built-up land and forest, respectively.

We note that Sentinel-2B was launched in 2017, whereas the CGLS-LC100 product was only available until 2019. Although we collect as much data (i.e., the Sentinel-2 AOD products and CGLS-LC100 products) as possible, results presented in the study are inevitably subject to statistical errors. Nevertheless, our results advance the understanding of the impacts of land cover types on atmospheric aerosols below city scales and facilitate land use decision making within a city. Future analyses should further focus on the roles of meteorological factors, topography, and landscape patterns on aerosols.

**Author Contributions:** Conceptualization, J.C.; methodology, Y.Y.; software, Y.Y. and K.Y.; writing—original draft preparation, Y.Y.; writing—review and editing, Y.Y. and J.C.; visualization, Y.Y., K.Y. and E.P.; funding acquisition, Y.C. All authors have read and agreed to the published version of the manuscript.

**Funding:** This research was funded by the China Scholarship Council (202006070089) and the Advance Research Project of Civil Aerospace Technology (D040402).

**Data Availability Statement:** Not applicable.

**Acknowledgments:** The Sentinel-2 AOD products are available from Yang et al. [10]. Sentinel-2 Level-1C (L1C) products are available from https://earthexplorer.usgs.gov/, accessed on 31 August 2021. CGLS-LC100 V3.0 products are available from https://lcviewer.vito.be/download, accessed on 24 November 2021. The authors wish to thank Babak Jahani for helpful suggestions. We thank the three anonymous reviewers for their careful reviews, which have helped improve the manuscript.

**Conflicts of Interest:** The authors declare no conflict of interest.

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
