# Peer review of "Land Use and Land Cover Influence on Sentinel-2 Aerosol Optical Depth below City Scales over Beijing"

_remotesensing, doi:10.3390/rs14184677_

Round 1
Reviewer 1 Report
The authors have analysed the Sentinel-2 AOD measurements over Beijing. They established the relationships between AOD and vegetation coverage, considering the normalized difference vegetation index and the normalized difference built-up index. The article is too short, and the results are not very original in terms of aerosols pollution and their origins. Nevertheless, the use of Sentinel-2 data is an interesting approach that merit publication. Perhaps the authors should increase the discussion part by better presenting the advantage of using the Sentinel-2 data. Also, they should better discuss which improvements in AOD analysis they have done in comparison with previous works on the same subject.
Specific comments:
The tile is too general. The authors must add “ Beijing » and « Sentinel-2 » in the title.
The authors must say in the abstract that the data were obtained by Sentinel-2.
Figure 1 is plotted thrice (the third one is in page 11)..
Figure 1 : Change « Corpland » to « Cropland » in the legend.
Lines 97-99: Some references must be given on the validation and the performance of the Sentinel-2 AOD products. I suggest to extend this part of the aper, showing the interest of using these data.
Line 140-142 : It is not necessary to write: “This section may be divided by subheadings. It should provide a concise and precise 140 description of the experimental results, their interpretation, as well as the experimental 141 conclusions that can be drawn.”
Line 224: Do you know what could be the “other factors”?
Line 256-257 : Are you sure that the dust storms are the only origin ? I think there is dust storms at other period of the year. Have you considered the possibility of spring agricultural activities and fertilizers? Also, have you considered the formation of secondary aerosols?
Line 110 is missing (masked by Figure 1!).
Lines 326-327 are unnecessary.
Reviewer 2 Report
This article investigates the relationship between five major land cover types and aerosol optical depth (AOD) in Beijing and its surroundings from 2017 to 2019. The authors showed that the patterns of aerosol distribution vary remarkably in time and space. While urban and built-up land is the major contributor to regional AOD, forest and grassland reduce AOD. They also concluded that anthropogenic activities have a non-negligible influence on AOD and can even reverse the contribution of certain land cover to aerosols.
General comments:
The article is generally easy to follow and the results are very interesting. However, the interpretation of the results is not convincing, at least to me, for example in Figure 5. The author divided the contour plot of NDBI vs NDVI colored by AOD into broadly three regions. This classification could be replaced by proper unsupervised classification, rather than dividing the regions manually. In Section 4.3, the authors used a threshold of 0.5 to divide the AOD groups. Are there any references for this value? Would the results be very different if other threshold (e.g. 0.4) is used? Therefore, I suggest major revision before publication.
Specific comments:
Ln 31: earth-atmosphere -> land-atmosphere
Ln 33: i.e. -> for example
Ln 77: Very vague to say ‘a small rectangular area’. It would be better to describe the actual area.
Figure 1: Cropland -> Cropland instead
Ln 109: download -> downloaded
Table 1: It would be clearer to include what the proportion of the land use in percentage.
Ln 140-142: Remove this.
Ln 159: Diving -> Dividing
Section 4: Discussion -> Results and Discussion
Figure 2: Put a legend label ‘AOD’
Ln 223: How does NDVI interact with NDBI? Can it be shown as a normalized ration of NDVI and NDBI?
Table 2: The results in 2018 look quite different from the rest. Any explanations for that?
Ln 326-327: Remove.
Reviewer 3 Report
This paper examined the relationships between AOD and land cover types, which is a reasonable and appropriate approach but requires a bit of modification, including data processing and accuracy.
L 91. Please describe the original resolution of S2 AOD. The authors should also explain why the NN method was used for downscaling the original S2 AOD to 100-m resolution.
L 99. Please present the accuracy of S2 and MODIS AOD using enough Aeronet ground measurements as reference data. Accurate AOD can support the results.
Round 2
Reviewer 1 Report
The authors have wellanswered to all my commentq. THus, the paper can be now publish.
Reviewer 3 Report
Well revised. Thank you for your efforts.